# Numerical Simulation of Crack Initiation and Propagation Evolution Law of Hydraulic Fracturing Holes in Coal Seams Considering Permeability Anisotropy and Damage

Liang Chen [1,2,3], Gangwei Fan [1,2], Dongsheng Zhang [1,2], Zhanglei Fan [2,*], Xufeng Wang [1,2], Wei Zhang [1,2] and Nan Yao [3]

1 State Key Laboratory of Coal Resources and Safe Mining, China University of Mining and Technology, Xuzhou 221116, China; chenliang_cumt@126.com (L.C.); fangw@cumt.edu.cn (G.F.); dshzhang123@126.com (D.Z.); wangxufeng@cumt.edu.cn (X.W.); zhangwei@cumt.edu.cn (W.Z.)
2 School of Mines, China University of Mining and Technology, Xuzhou 221116, China
3 Hubei Key Laboratory for Efficient Utilization and Agglomeration of Metallurgic Mineral Resources, Wuhan 430081, China; yaonan@wust.edu.cn
* Correspondence: tb18020005b4@cumt.edu.cn

**Abstract:** Hydraulic fracturing has been widely used in practical engineering as an essential means to prevent coal seam gas outburst, increase coal seam permeability and improve gas drainage efficiency. Accurate prediction of fracture propagation law is an important basis for optimizing fracturing parameters to achieve high-efficiency gas drainage in coal seams. In this paper, a new seepage–stress–damage coupling model considering permeability anisotropy is first established and then used to study the evolution laws of crack initiation pressure ($\sigma_{ci}$), fracture pressure ($\sigma_{cd}$), AE behavior and pore water pressure with the lateral pressure coefficient ($\xi$) and permeability anisotropy coefficient ($\lambda$) in the process of hydraulic fracturing. Finally, the influence of initial pore water pressure on $\sigma_{ci}$ is discussed, and an efficient gas drainage method is proposed. Research results indicate that: the in situ stress still plays a leading role in the approach of crack propagation whether the permeability is isotropic or anisotropic; the non-uniform pressure condition is favorable for the crack growth compared with uniform pressure under the isotropic permeability condition; when the direction of maximum permeability is consistent with the direction of maximum principal stress ($\xi = 0.5$, $\lambda < 0$), the coal seams are easily fractured; AE behavior of fracturing holes can be divided into three stages: initiation stage, fracture smooth expansion stage and the breakdown stage for any $\lambda$ or $\xi$; and the more complex the crack distribution, the more the area of the gas pressure release zone (GPRZ) increases, which is very beneficial to achieve high-efficiency gas drainage. This study can provide a basis for optimizing fracturing parameters and technology in improving the efficiency of coal seam gas drainage using the hydraulic fracturing method.

**Keywords:** hydraulic fracturing; crack propagation; AE behavior; cumulative gas extraction volume



## 1. Introduction

With the advent of the post-oil era, unconventional oil and gas resources have become more and more important in the global energy structure [1]. In order to further enhance oil and gas recovery, a large number of permeability-increasing and production-increasing technologies have been proposed. Nonetheless, all kinds of technologies have inherent applicability and limitations [2,3]. In recent years, as an effective and mature technology, hydraulic fracturing has been widely used in the gas drainage of coal seams [4]. The improvement in gas drainage effect of coalbed methane reservoirs by hydraulic fracturing is closely related to fracturing borehole layout, in situ stress conditions, coal seam permeability and other parameters [5]. As a result of unreasonable fracturing parameter setting, it is challenging to achieve the gas drainage effect, and it even causes a series of problems

such as gas outburst and groundwater pollution [6,7]. The establishment of an accurate mathematical model and reasonable numerical solution to study the crack propagation law and gas drainage effect after fracturing under different geological conditions is widely used in this field, and can provide a basic reference for the evaluation of the feasibility and effectiveness of improving the gas drainage effect by the pre-fracturing method and the layout selection of drilling parameters.

In order to effectively reveal the crack evolution mechanism during fracturing, different scholars have carried out a large number of studies on rock nonlinear damage, crack initiation, propagation and penetration mechanisms [8]. Wang and Watanabe et al. established a large number of analytical and semi-analytical solutions based on tensile stress and stress intensity to predict crack initiation pressure [9–11]. Then, Zhang, Solberg and Estrada found that the rock initiation pressure was related to fracturing fluid properties and injection rates [12–14]. To explain these phenomena, ITO and Hayashi [15,16] proposed a new model based on the point stress criterion and applied it to reveal the relationship between fluid injection parameters and crack initiation and propagation. In recent years, discrete element, continuous element, discrete-continuous mixed element and phase field methods have been widely used to accurately simulate the law of fracture propagation in hydraulic fracturing [17,18]. Zhu et al. [19] established a mechanical calculation model of crack initiation pressure considering the in situ stress and fluid–solid coupling. Zhao et al. [20] discussed the crack propagation law under high water pressure based on the self-developed seepage-fracture coupling numerical calculation method. He et al. [21] discussed how to use discrete element software to accurately characterize the fracture morphology in the process of hydraulic fracturing, but this cannot overcome the defects of the software itself (tensile cracks lead to joint enlargement, which may cause joint disappearance, and then lead to a complex fracture network). Fan et al. [22] established a thermal-fluid–solid coupling damage model of heterogeneous rock considering non-Darcy effects to simulate the hydraulic fracturing process, and revealed the influence of non-Darcy effects on the extraction yield. Al-Rubaie and Mahmud [23] investigated the performance of hydraulic fracturing in naturally fractured gas reservoirs based on the stimulated rock volume using the DEM method. Wang et al. [24] studied the hydraulic fracture propagation and interaction with discontinuous natural fracture networks in coal seams based on the cohesive element method.

In the above research, the rock–coal seam is regarded as the permeability isotropic body. However, in practical engineering, the surrounding rock often shows obvious anisotropic characteristics under the influence of joints, cracks or bedding, especially for coal seams. Furthermore, in the process of hydraulic fracturing, the coal around the fracturing hole is obviously damaged by the weakening effect of water erosion, and the damaged coal mass is more likely to crack under the pore pressure. Therefore, comprehensively considering the influence of permeability anisotropy and damage effects, the study of the crack propagation law and gas drainage effects in the process of hydraulic fracturing under different geological conditions can be closer to engineering practice.

This paper first establishes a new seepage–stress–damage coupling model considering permeability anisotropy, and then its accuracy is verified compared with the experimental results. The model was embedded into COMSOL software to study the evolution laws of crack initiation pressure ($\sigma_{ci}$), fracture pressure ($\sigma_{cd}$) and pore pressure with the lateral pressure coefficient ($\xi$) and permeability anisotropy coefficient ($\lambda$) during the hydraulic fracturing. Finally, the influence mechanism of permeability anisotropy and pore water pressure on $\sigma_{ci}$ is discussed, and an efficient gas drainage method is proposed by using special graded particles to seal cracks. It should be noted that the $\sigma_{ci}$ is initial water pressure when the crack is first generated and $\sigma_{cd}$ is the water pressure corresponding to the crack rapid expansion; both can be determined by the numerical simulation result [16].

## 2. Establishment of Seepage–Stress–Damage Coupling Model with Permeability Anisotropy

### 2.1. Basic Assumption

According to the physical and mechanical properties of coal seams, the gas occurrence environment and the migration mechanism, combined with previous research results, the following assumptions are proposed [25,26]: ① The coal mass is a kind of elastic continuum with a single pore structure and single permeability; ② the crack is saturated by free water, free gas and adsorbed gas, which meet the requirements of Darcy's seepage law; ③ the migration of water and gas in cracks is an isothermal migration process; ④ the coal mass belongs to a permeability anisotropic medium; and ⑤ the fissure in the process of hydraulic fracturing can be indirectly characterized by damage. The yield of the mode element obeys the Mohr–Coulomb criterion and strength parameters obey the Weibull distribution: $f(u) = m/u_0 \ (u/u_0)^{m-1} \exp [-(u/u_0)^m]$; $u_0$ is the average value of mechanical parameters of elements; $m$ is the homogeneity index.

### 2.2. Governing Equation of Seepage Field

According to the assumption of porous media, fractured water, groundwater and gas are present in a coal seam during hydraulic fracturing. The fractured water is generated by the input of hydraulic fracturing system into the coal seam. When the water pressure is applied, the fractured water and groundwater in the coal seam are collectively referred to as high-pressure water. Taking the high-pressure water and the formed fracture channel as the power and path of fluid migration, respectively, the dynamic equilibrium state of gas is broken after the initial adsorption/desorption, and its seepage equation can be expressed as [27]:

$$\frac{\partial}{\partial t}\left(s_g \varphi \frac{M_g}{RT} p_g\right) + \frac{\partial}{\partial t}\left(\frac{V_L p_g}{P_L + p_g}\rho_c \rho_{gs}\right) + \nabla \cdot \left(-\frac{k k_{rg}}{\mu_g}\left(1 + \frac{b}{p_g}\right)\frac{M_g}{RT} p_g \nabla p_g\right) = 0 \quad (1)$$

where $\varphi$ is porosity; $\rho_c$ is the density of coal, kg/m$^3$; $\rho_{gs}$ is the gas density under standard conditions, kg/m$^3$; $M_g$ is the molar mass of gas, kg/mol; $R$ is the molar constant of gas, J/(mol·K); $p_g$ is gas pressure, MPa; $T$ is the coal seam temperature, K; $V_L$ is the Langmuir volume constant, m$^3$/kg; $P_L$ is Langmuir pressure constant, Pa; $k$ is absolute permeability, m$^2$; $k_{rg}$ is gas phase permeability; $\mu_g$ is the dynamic viscosity of gas, Pa·s; $b$ is the slippage factor, Pa.

The water transport equation obeys Darcy's law. The water transport equation reflecting the gas–water two-phase flow with saturation as a variable can be expressed as follows [28]:

$$\frac{\partial(s_w \phi \rho_w)}{\partial t} + \nabla \cdot \left(\rho_w - \frac{k k_{rw}}{\mu_w}\nabla p_w\right) = 0 \quad (2)$$

where $s_w$ is water phase saturation; $\rho_w$ is water density, kg/m$^3$; $k_{rw}$ is the relative permeability of water phase; $\mu_w$ is the dynamic viscosity of water phase, Pa·s; $p_w$ is the water pressure in the fracture, MPa.

There are many empirical expressions of relative permeability as a comprehensive reflection of gas–water two-phase flow seepage characteristics. Based on the capillary pressure curve, this paper adopts a more commonly used form as follows [29]:

$$\begin{cases} k_{rg} = k_{rg0}\left(1 - \left(\frac{s_w - s_{wr}}{1 - s_{wr} - s_{gr}}\right)\right)^2 \left(1 - \left(\frac{s_w - s_{wr}}{1 - s_{wr}}\right)^2\right) \\ k_{rw} = k_{rw0}\left(\frac{s_w - s_{wr}}{1 - s_{wr}}\right)^4 \end{cases} \quad (3)$$

where $s_{wr}$ is irreducible water saturation, 0.42; $s_{gr}$ is residual gas saturation; $k_{rg0}$ is the relative permeability of gas phase endpoint, 0.756; $k_{rw0}$ is the relative permeability of water phase endpoint.

The governing equation of hydraulic fracturing seepage field can be obtained by combining Equations (1)–(3):

$$\begin{cases} \frac{\partial}{\partial t}(s_w\phi\rho_w) + \nabla \cdot \left[-\rho_w\frac{kk_{rw0}}{\mu_w}\left(\frac{s_w-s_{wr}}{1-s_{wr}}\right)^4\nabla p_w\right] = 0 \\ \frac{\partial}{\partial t}\left(s_g\phi\frac{M_g}{RT}p_g\right) + \frac{\partial}{\partial t}\left(\frac{V_Lp_g}{P_L+p_g}\rho_s\rho_{gs}\right) + \nabla \cdot \left(-\frac{kk_{rg0}}{\mu_g}\left(1-\left(\frac{s_w-s_{wr}}{1-s_{wr}-s_{gr}}\right)\right)^2\left(1-\left(\frac{s_w-s_{wr}}{1-s_{wr}}\right)^2\right)\left(1+\frac{b}{p_g}\right)\frac{M_g}{RT}p_g\nabla p_g\right) = 0 \end{cases} \quad (4)$$

### 2.3. Equation of Solid Stress Field

According to the generalized Hooke's law, the strain contribution term of porous media can be divided into the stress term, the fluid pressure term and the gas adsorption/desorption strain term. Combined with the geometric relationship and static equilibrium relationship of coal mass under small deformation conditions, the stress field governing equation considering pore fluid pressure, gas adsorption and damage effect (reflected by elastic modulus attenuation) can be determined [30]:

$$Gu_{i,jj} + \frac{G}{1-2v}u_{j,ji} - \alpha p_{f,i} - K\varepsilon_{a,i} + F_i = 0 \quad (5)$$

In addition:

$$\begin{cases} p_f = s_w p_w + s_g p_g \\ p_w = p_g - p_{cgw} \\ G = E/2(1+v) \\ K = E/3(1-2v) \\ E = E_0(1-D) \end{cases}$$

where $G$, $K$ and $E$ are shear modulus, bulk modulus and elastic modulus of coal, respectively, Pa; $v$ is Poisson's ratio; $\alpha$ is Biot coefficient; $p_f$ is the fluid pressure in the fracture, Pa; $\varepsilon_a$ is the adsorption strain; $F_i$ is the volume force, Pa; the symbol $(i, j)$ corresponds to the coordinates $(x, y)$; $p_{cgw}$ is capillary pressure, MPa; $D$ is the damage variable; $E_0$ is the initial elastic modulus, Pa.

### 2.4. Governing Equation of Damage Field

During hydraulic fracturing, due to the stress field distribution differences and heterogeneity of rock material, different damage fracture zones are formed at different locations around boreholes. The stress state of surrounding rock around the borehole can be determined by the maximum tensile failure criterion and the Mohr–Coulomb criterion (the stress follows the principle of positive tension and negative pressure) [31]:

$$\begin{cases} F_1 \equiv \sigma_1 - f_{t0} \\ F_2 \equiv -\sigma_3 + \sigma_1[(1+\sin\theta)/(1-\sin\theta)] - 2C/(1-sin\theta) \end{cases} \quad (6)$$

where $f_{t0}$ is tensile strength, Pa; $C$ is cohesion of coal, which can be converted from uniaxial compressive strength, MPa; $\theta$ is the internal friction angle of coal; $F_1$ and $F_2$ are functions of stress state, $\sigma_1$ and $\sigma_3$ are the first and third principal stresses respectively. The damage variable can be defined as:

$$D = \begin{cases} 0 & F_1 < 0 \text{ and } F_2 < 0 \\ 1 - \left|\frac{f_{t0}}{E\varepsilon_1}\right|^n & F_1 = 0 \text{ and } dF_2 > 0 \\ 1 - \left|\frac{2C}{E(1-sin\theta)\varepsilon_3}\right|^n & F_2 = 0 \text{ and } dF_1 > 0 \end{cases} \quad (7)$$

where $n$ is a constant representing the brittle-plastic properties of rock after fracture, and can be obtained by fitting the stress–strain curve.

### 2.5. Definition of Permeability

Porosity and permeability are the critical parameters in the hydraulic fracturing process. Considering the effect of permeability directionality on stress and seepage fields, the permeability can be expressed as [32]:

$$k_i = k_{i0}\left[1 - \frac{1}{\phi_0 + 3K_f/K}\left[(\varepsilon_a - \varepsilon_{a0}) - 3(\varepsilon_j - \varepsilon_{j0})\right]\right]^3, i \neq j \tag{8}$$

where $K_f$ is the improved crack stiffness, $\varphi_0$ is the initial porosity; $\varepsilon_{a0}$ and $\varepsilon_a$ are the initial and current adsorption strains respectively. $k_{i0}$ and $k_i$ are initial permeability and current permeability in different directions of the coal seam; $\varepsilon_i$ is the strain in different directions $(i = x, y)$.

The permeability of coal seams increases significantly after hydraulic fracturing, and then the permeability can be expressed as:

$$k_{wi} = k_i \exp(D \times ak) \tag{9}$$

where *ak* is the permeability jump coefficient.

The stress–seepage–damage coupling model of coal seam in the process of hydraulic fracturing considering permeability anisotropy is established by combining Formulas (4), (5), (7) and (9). The coupling relation is shown in Figure 1. The established fluid–solid coupling model was programmed in MATLAB, and then the MATLAB program was embedded in COMSOL software to carry out the following research. The simulation calculation steps are shown in Figure 2. In the calculation process, the damage is used as a criterion to correct the stress state of the unit. If the damage occurs, the elastic modulus of the unit is updated, the stress field distribution is recalculated and the stress state around the borehole is continuously updated until no new damage occurs. Then, the time step is increased, and the exact cycle damage identification is carried out until the fracturing end.

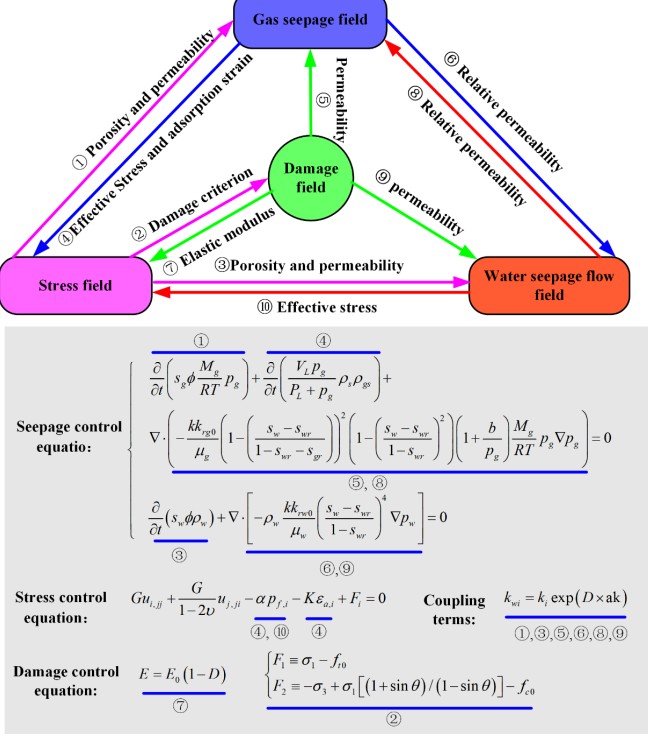

**Figure 1.** Coupling relations of different physical fields.

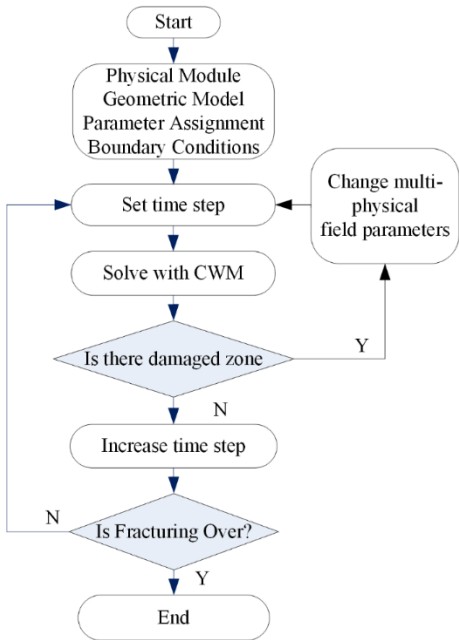

**Figure 2.** Solving process of the coupled damage–stress–seepage model of coal seams.

## 3. Evolution Law of Crack Propagation under Different Geological Conditions

### 3.1. Model Validation

For verifying the correctness of the stress–seepage–damage coupling model and the numerical iterative calculation process, two methods were used to verify the model: (1) the seepage characteristics of standard specimens under triaxial compression; (2) comparison between the numerical solution and theoretical solution of initiation pressure of hydraulic fracturing boreholes.

Figure 3 presents the seepage test model of the standard specimen under the triaxial compression established by this simulation. The boundary conditions, rock parameters and seepage parameters imposed by this model are basically consistent with the test conditions in reference [33]; that is, the top of the specimen is loaded at the speed of 0.05 m/s, the pore pressures at the top and bottom of the specimen are 1 and 0.5 MPa, respectively, and the confining pressure of 2 MPa is applied on both sides.

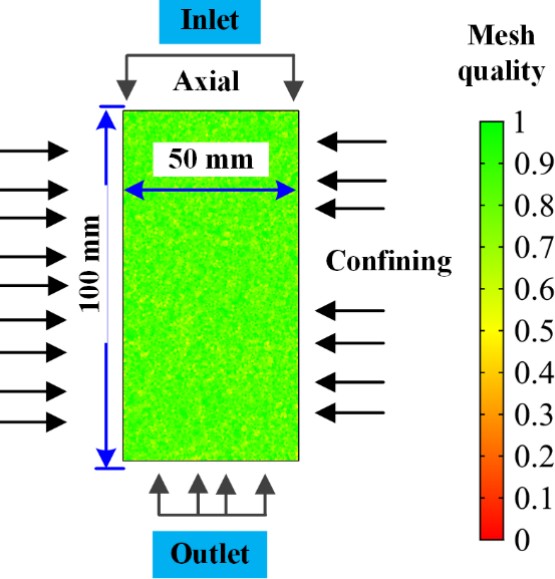

**Figure 3.** Numerical calculation model of triaxial compression.

Figure 4 shows the comparison of numerical simulation results and experimental results of stress and permeability. It can be seen from Figure 4 that the permeability of specimens at the initial stage of loading (before the B state point) decreases slightly with the increasing stress regardless of the numerical simulation or the test results. Although there is a local damage area inside the specimen, its distribution is relatively isolated and has little effect on the overall permeability of specimens. When the strain exceeds the strain state point corresponding to point B, the permeability begins to increase slowly at first and then increases rapidly, and reaches the maximum value at the post-peak stage (D state point). At this time, the specimen has a macro fracture that runs through the upper and lower ends. As the main seepage channel, the fracture leads to a substantial increase in permeability. In summary, the stress−strain curve and permeability−strain curve obtained by the numerical simulation are basically consistent with the experimental results in reference [33], which verifies the correctness of the model established in this paper.

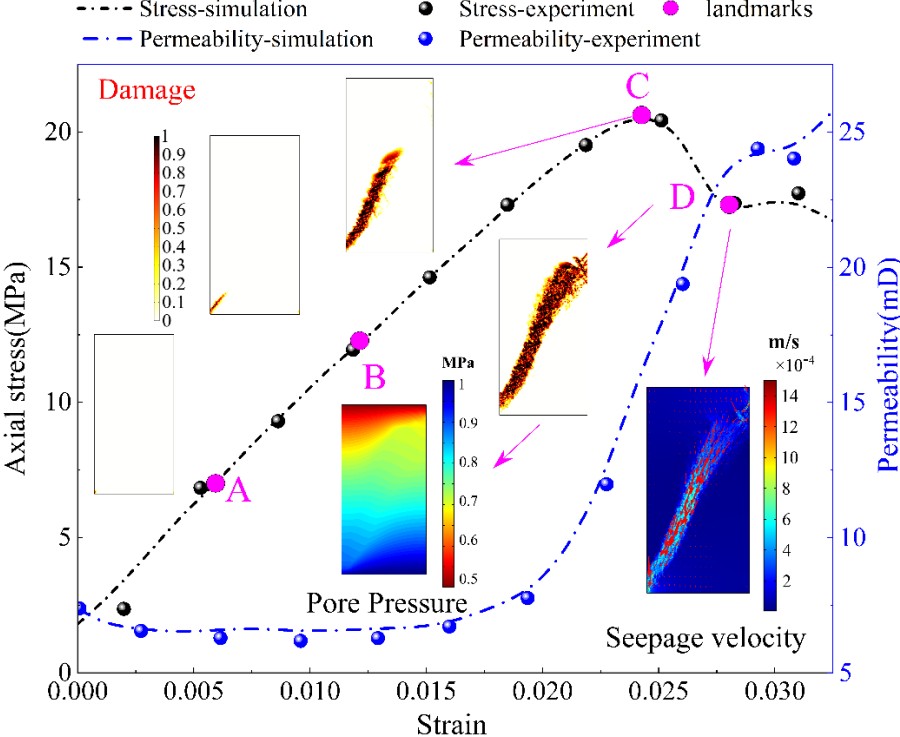

**Figure 4.** Comparison between the simulation and experimental results of stress−strain and permeability−strain curve.

In order to further verify the accuracy of the model, the numerical calculation model of hydraulic fracturing boreholes was established, as shown in Figure 5, where the vertical stress is 10 MPa and the horizontal stress is 5~20 MPa. The parameter ($\xi$) is used to characterize the different in situ stress conditions. The left and lower boundaries of the model are set as roller shaft support, and the Delaunay method is used to mesh. The degree of freedom is 573,684. During the numerical calculation, the pore pressure of the fracturing borehole increases by 0.25 MPa/step, 50 cycles per step, and the number of calculation steps 2–50 represents the 50th cycle in the second calculation step. By changing the horizontal stress, multiple sets of numerical simulation experiments were carried out. Finally, the numerical calculation results of the initiation stress were compared with those of the classical theoretical model. The numerical simulation parameters are shown in Table 1.

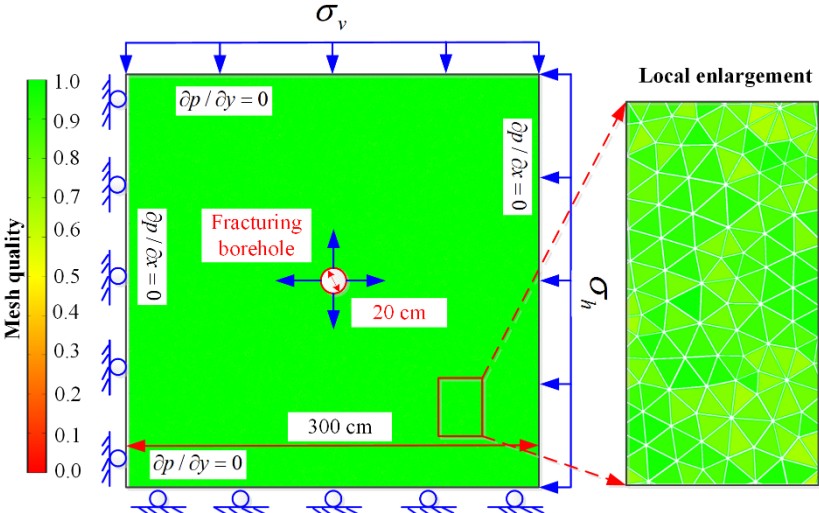

**Figure 5.** Numerical calculation model of hydraulic fracturing and boundary conditions.

**Table 1.** Physical and mechanics parameters of numerical calculation model.

| Parameters | Value | Parameters | Value | Parameters | Value |
|---|---|---|---|---|---|
| $E$ | 10 GPa | $k_0$ | $1 \times 10^{-17}$ m$^2$ | $ak$ | 6 |
| $v$ | 0.3 | $p_0$ | 0 MPa | $b$ | 0.76 MPa |
| $f_{t0}$ | 1.4 MPa | $\mu_{w}$ | $10^{-3}$ Pa s | $\mu_g$ | $1.84 \times 10^{-5}$ Pa·s |
| $m$ | 10 | $\alpha$ | 0.9 | $p_{cgw}$ | 0.05 MPa |

Note that: $m$ is homogeneity coefficient; $k_0$ is initial permeability; $p_0$ is initial water pressure.

Other parameters are introduced in Section 2.

The Hubbert–Willis (H-W) [34] formula and Haimson–Fairhurst (H-F) [35] formula are the main theoretical formulas for initiation pressure calculation. The H-W formula is suitable for non-permeable rock, and the H-F formula is suitable for permeable rock. The specific expressions are:

$$\begin{cases} p_{H-W} = 3\sigma_3 - \sigma_1 - p_0 + f_t \\ p_{H-F} = \dfrac{3\sigma_3 - \sigma_1 - p_0 + f_t}{2 - \alpha(1-2v)/(1-v)} - p_0 \end{cases} \tag{10}$$

The numerical calculation solutions ($v$ = 0.3, $\alpha$ = 0.9) and theoretical solutions of initiation pressure under different lateral pressure coefficients, Poisson's ratios and Biot coefficients are shown in Figure 6, where the simulation results under the above different parameters present good consistency, which is closer to the theoretical solution of the permeable rock. With the increasing parameter ($\xi$), the initiation pressure of fracturing boreholes increases first and then decreases, and reaches the maximum when $\xi$ = 1. This further confirms the correctness and rationality of the model in this paper, indicating that the model can be used to quantitatively study the crack initiation and propagation behavior of hydraulic fracturing boreholes in heterogeneous coal masses under different in situ stress conditions.

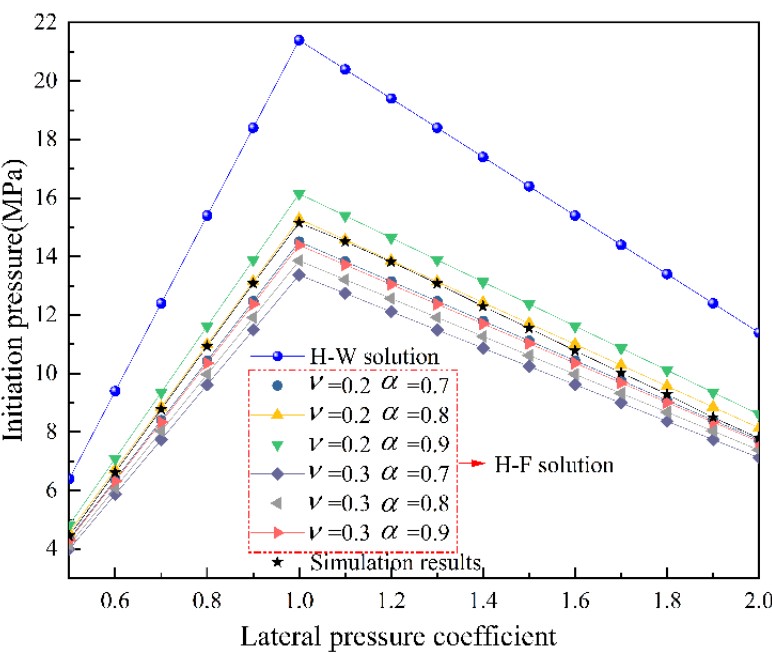

**Figure 6.** Comparison between numerical simulation and theoretical calculation results of crack initiation pressure.

*3.2. Crack Propagation Law under Different In-Situ Stress Condzitions*

To investigate the effect of in situ stress conditions on the crack propagation law, the crack evolution process around the hydraulic fracturing drilling was simulated and analyzed under two typical ground stress conditions ($\xi$ = 0.5, 1). The crack expansion, damage zone expansion, pore pressure distribution and degradation of elastic modulus under different lateral pressure coefficients and fracturing steps are shown in Figures 7 and 8. It can be seen from Figures 7 and 8 that the crack propagation process under the two lateral pressure coefficients can be roughly divided into three stages: the initiation stage, the fracture smooth expansion stage, and the breakdown stage. The crack evolution characteristics and pore pressure distribution around fracturing holes in each stage can be summarized as follows:

① Under different lateral pressure coefficients, the crack initiation stage of the fracturing hole is relatively similar. At this stage, there are few damage points around the boreholes, and their distribution is uneven. The shape of the water pressure distribution is close to the circle (Figure 7c), and the crack propagation speed is low, or even does not extend further, indicating that the coal seam cannot be damaged continuously after the crack initiation.

② After the crack propagation enters the fracture smooth expansion stage, the influence of lateral pressure coefficient on the crack propagation direction and process begins to appear. When $\xi$ = 0.5, the maximum principal stress is in the vertical direction, and the crack extends intermittently and slowly along the direction of the maximum principal stress, and the borehole water pressure has an oval-like distribution (Figure 7c). When $\xi$ = 1.0, the properties of fracturing materials play a leading role in the crack propagation, the force transmission among internal particles is not continuous, and the deformation of internal particles is not coordinated, resulting in many fractures, tortuous propagation paths and large randomness of propagation direction. At this time, the pore pressure distribution is irregular (Figure 8b).

③ After the crack development enters the breakdown stage, the branch cracks begin to gather and expand along the fixed direction. The crack evolves from the slender shape to the wide and disorderly shape, and the degradation of elastic modulus is consistent with the crack evolution (Figures 7d and 8c).

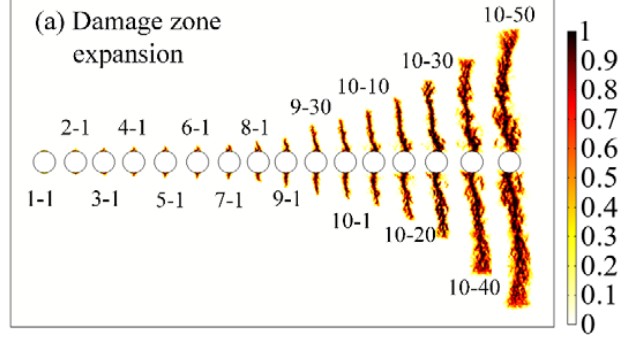

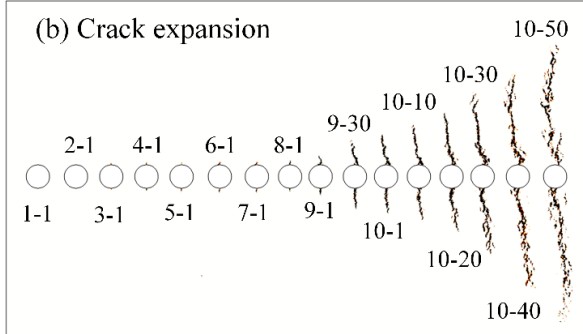

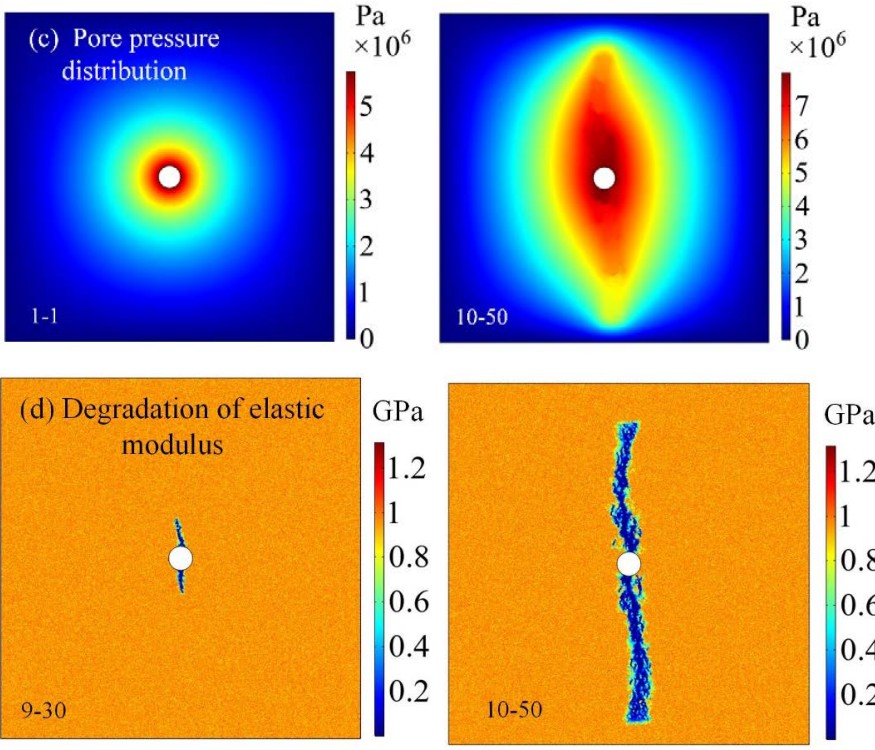

**Figure 7.** Damage zone expansion, crack expansion, pore pressure distribution and elastic modulus degradation law around hydraulic fracturing drilling when $\xi = 0.5$. (Note that: the color bar in (**a**) represents the damage variable, *D*; the color bar in (**b**) represents the crack expansion, (**c**) represents the pore pressure value; the color bar in (**d**) represents the elastic modulus value at different areas).

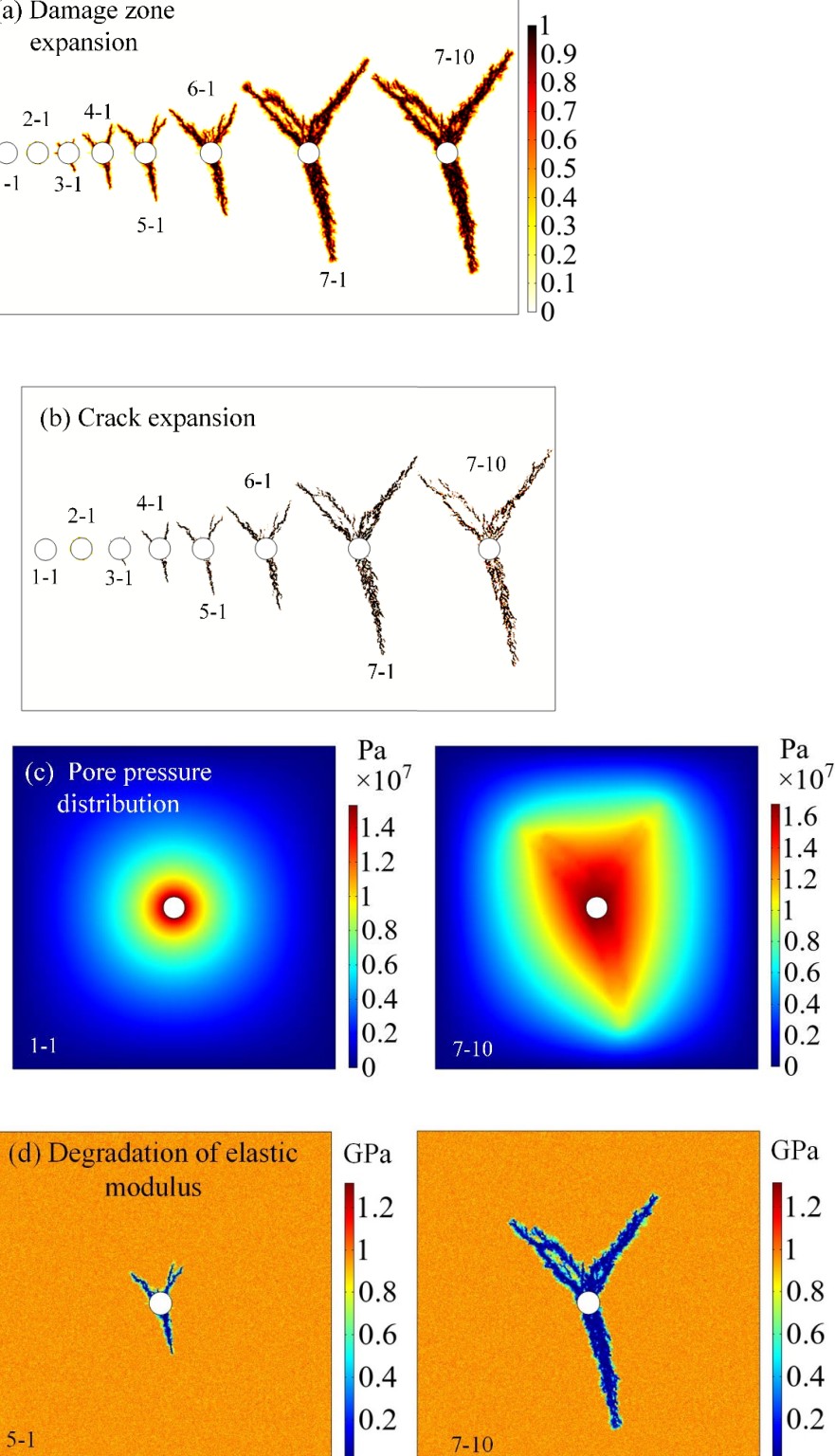

**Figure 8.** Damage zone expansion, crack expansion, pore pressure distribution and elastic modulus degradation law around hydraulic fracturing drilling when $\xi = 1.0$. (Note that: the color bar in Figure 8 (**a**) represents the damage variable, *D*; the color bar in Figure 8 (**b**) represents the crack expansion, (**c**) represents the pore pressure value; the color bar in Figure 8 (**d**) represents the elastic modulus value at different areas.).

Figure 9 shows the evolution law of the cumulative crack length, damage zone areas, and cumulative acoustic emission (AE) counts with the calculation time step at different hydraulic fracturing stages. As can be seen from Figure 9, the evolution characteristics of the cumulative crack length, the growth speed of damage zone areas, and the growth rate of cumulative AE counts with the calculation time step obtained by this simulation are well-matched with those at the above three stages. The micro-crack propagation process under different lateral pressure coefficients is basically the same, and the initiation pressure is 4.45 MPa for $\xi = 0.5$ and 15.15 MPa for $\xi = 1$. When the fracturing process enters the fracture smooth expansion stage, the damage zone area and cumulative AE counts start to gradually increase, of which the increased amplitude under $\xi = 1$ is significantly greater than that under $\xi = 0.5$.

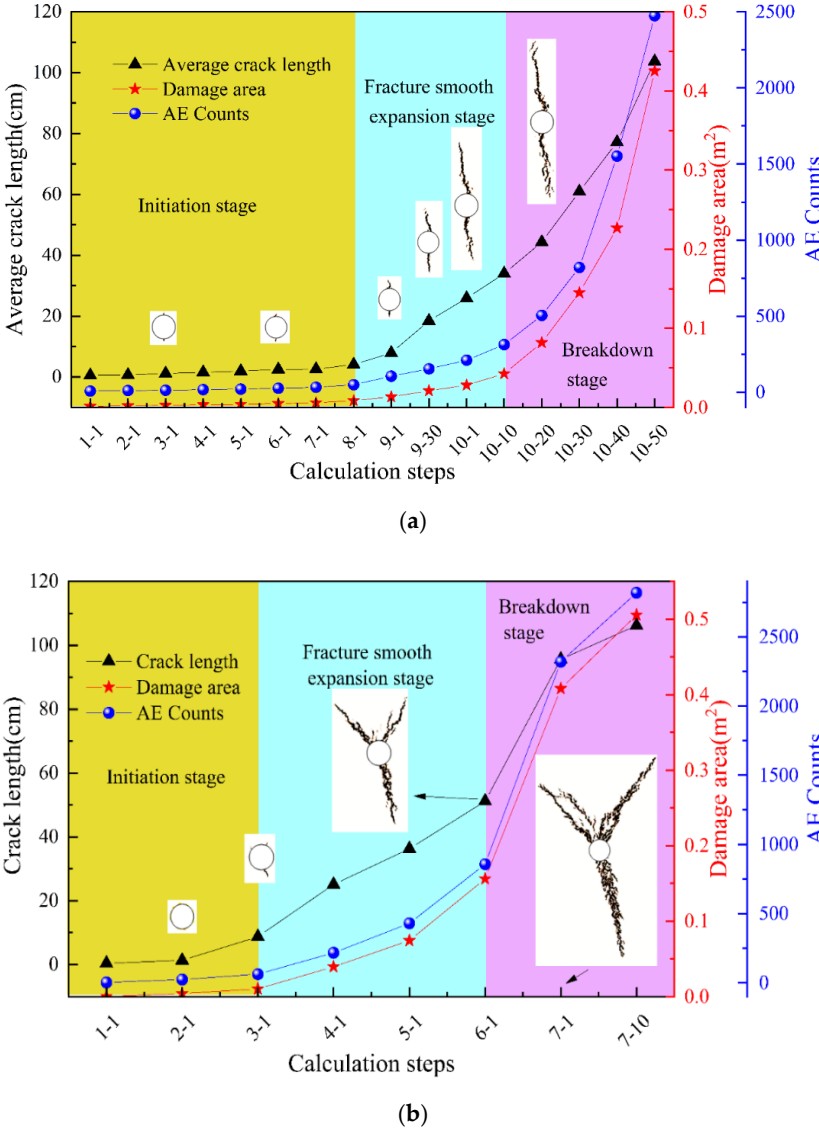

(a)

(b)

**Figure 9.** Cumulative crack length, damage zone area and cumulative AE counts of fracturing hole change with calculation time step at different hydraulic fracturing stages: (**a**) $\xi = 0.5$; (**b**) $\xi = 1.0$.

In addition, regardless of the lateral pressure coefficients, the AE parameters at the fracture stage increase rapidly, accompanied by a large number of unit damages. Furthermore, the fracture pressure is 8.25 MPa for $\xi = 0.5$ and 17.05 MPa for $\xi = 1$. The above results show that the uniform pressure condition is not conducive to the crack propagation

around fracturing holes compared with the non-uniform pressure condition, but has a significant influence on its fracture degrees.

### 3.3. Crack Propagation Law under Permeability Anisotropy

The degree of permeability heterogeneity anisotropy is closely related to the internal pore structure, joint distribution and stress environment of coal masses, expressed as the permeability anisotropy coefficient ($\lambda$), which is the ratio of horizontal permeability to vertical permeability. In this section, taking $\xi = 1.0$ when $\lambda$ is 0.1 and 0.5, respectively, the influence of $\lambda$ on crack propagation and pore pressure variation is studied. Figure 10 shows the progressive evolution law of cracks and final pore pressure distribution around the fracturing hole at various values of $\lambda$. Figure 11 indicates the cumulative crack lengths and AE characteristic around the fracturing hole at different fracturing steps. Table 2 shows the variation in crack initiation pressure, fracture pressure and their ratio at various $\lambda$. It can be seen from Figures 10 and 11, and Table 2, that:

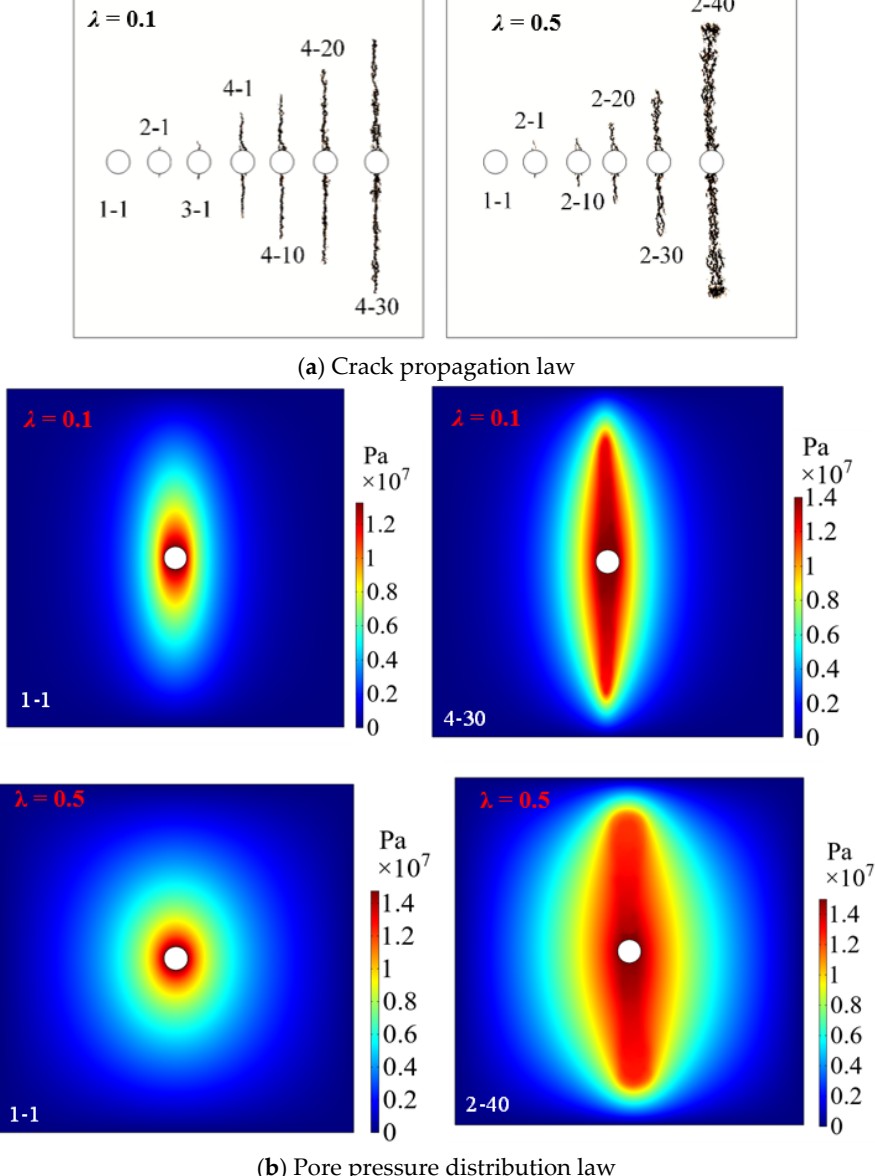

**Figure 10.** Crack propagation and pore pressure distribution law under different $\lambda$.

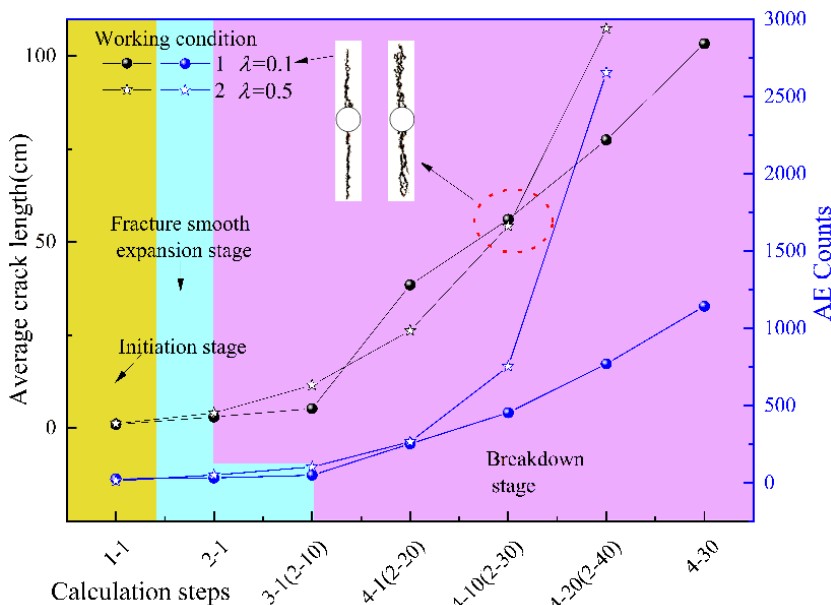

**Figure 11.** Cumulative crack length and AE counts of fracturing hole change with calculation time step under different $\lambda$ when $\xi$ = 1.0.

**Table 2.** Crack initiation stress, fracture stress and their ratio under different $\lambda$.

| $\lambda$ | $\sigma_{ci}$ (MPa) | $\sigma_{cd}$ (MPa) | $\sigma_{ci}/\sigma_{cd}$ (%) |
| --- | --- | --- | --- |
| 0.1 | 13.25 | 14.0 | 94.64 |
| 0.5 | 14.6 | 15.00 | 97.33 |
| 1.0 | 15.15 | 17.05 | 88.86 |

① Permeability anisotropy has significant influences on crack development and the pore pressure distribution of fracturing holes. Under the effect of permeability anisotropy, the initial pore pressure is elliptically distributed, and the ratio of the short side length to the long side length is basically equal to its corresponding anisotropy coefficient; the crack propagation direction of fracturing holes basically extends along the maximum permeability direction. The larger the anisotropy coefficient, the stronger the integrity of fracture propagation, and the fewer the branches.

② When the anisotropy coefficient is respectively 0.1 and 0.5, the corresponding initiation pressures are slightly lower than that under the isotropy condition (15.15 MPa), which are 13.25 and 14.6 MPa, respectively, with a decrease in amplitude of 12.54% and 3.63%; the decrease in amplitude of fracture pressure is 17.9% for $\lambda$ = 0.1 and 12.0% for $\lambda$ = 0.5. The above analysis indicates that the permeability anisotropy will reduce the initiation pressure and fracture pressure under the isotropic in situ stress condition, and the reduction in fracture pressure is greater than that of the initiation pressure.

③ The crack propagation process and AE behavior under different $\lambda$ can also be divided into three stages: the initiation stage, the fracture smooth expansion stage and the breakdown stage. In the initiation stage, the cumulative crack length and AE counts under $\lambda$ = 0.1 and $\lambda$ = 0.5 are roughly equal. However, when the fracturing process enters the fracture smooth expansion stage, the above two parameters under $\lambda$ = 0.5 are slightly greater than those under $\lambda$ = 0.1. Once the fracturing process begins to enter the breakdown stage, the above two parameters under $\lambda$ = 0.5 are significantly greater than those under $\lambda$ = 0.1, indicating that the higher the permeability anisotropy degree, the higher the degree of crack development, and the more significant the hydraulic fracturing effect.

### 3.4. Crack Propagation Law of Fracturing Holes Considering the Coupling Effect of ξ and λ

The relationship between maximum ξ and λ direction is also an important factor affecting the crack propagation and AE behavior of fracturing holes. Figure 12 shows the crack evolution law of fracturing holes under different λ when ξ = 0.5. Figure 13 indicates the change in cumulative crack length and AE counts of fracturing holes with the calculation time step under different λ when ξ = 0.5. Furthermore, the variation law of crack initiation pressure, fracture pressure and their variation amplitude with the parameter λ when ξ = 0.5 is also presented in Figure 14. As can be seen from Figures 12–14:

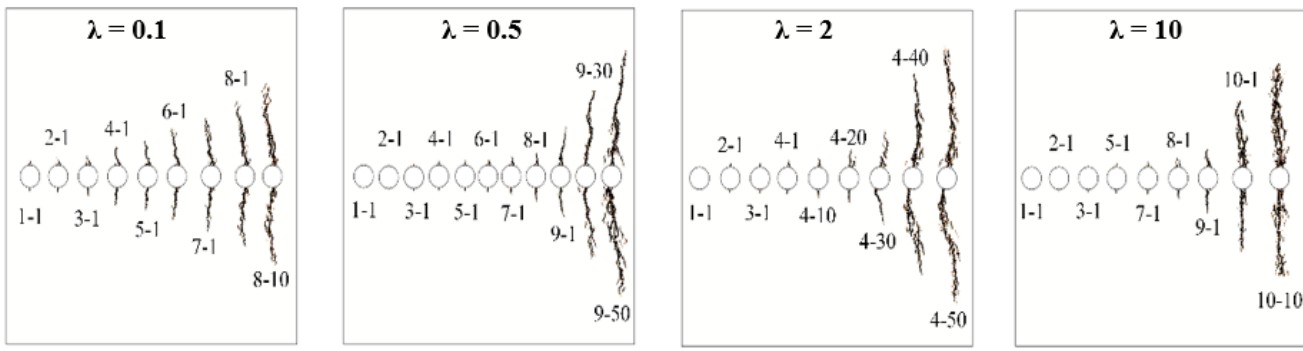

**Figure 12.** Crack propagation law of fracturing hole under different λ when ξ = 0.5.

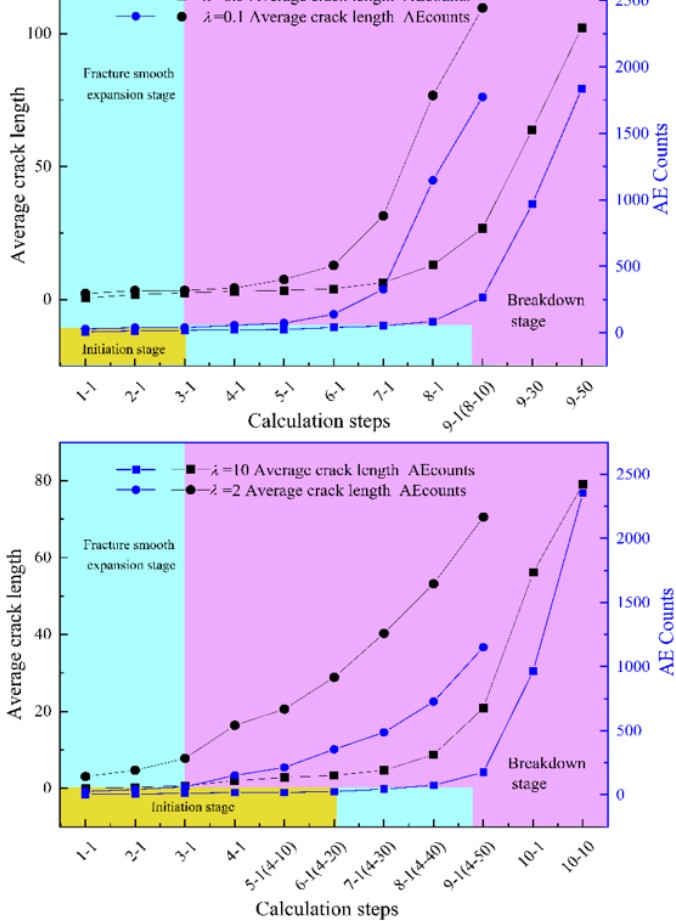

**Figure 13.** Cumulative crack length and AE counts of fracturing hole change with calculation time step under different λ when ξ = 0.5.

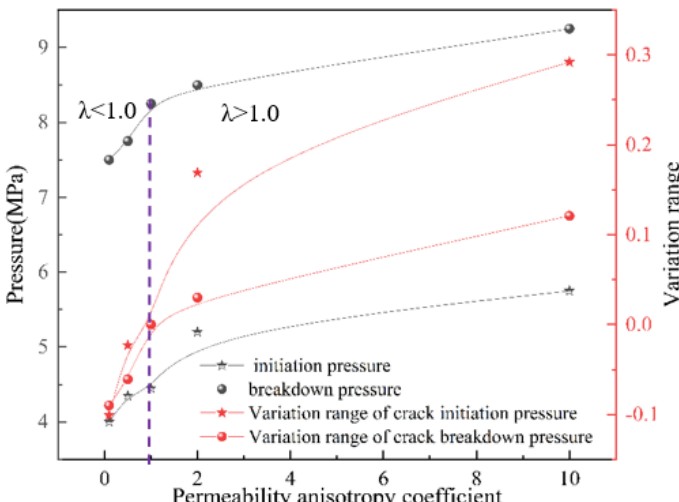

**Figure 14.** The variation law of crack initiation stress, fracture stress and their variation amplitude with the permeability anisotropy coefficient when $\xi = 0.5$.

① When $\xi = 0.5$, regardless of the largest permeability in the horizontal or vertical direction, the crack of fracturing holes propagated along the vertical direction, indicating that the in situ stress still plays a dominant role in the direction of crack propagation, which may be the reason why predecessors only consider the in situ stress when studying the direction of crack propagation [34,35].

② The relationship between maximum $\lambda$ and $\xi$ direction has a significant impact on the initiation and fracture pressures of the fracturing hole. When the direction of maximum permeability is consistent with the direction of maximum principal stress ($\xi = 0.5$, $\lambda < 0$), the initiation and fracture pressures of the fracturing hole obviously decrease with the decreasing $\lambda$. The decreasing rate also increases significantly with the decrease in $\lambda$, indicating that the higher permeability directionality is more favorable to the crack initiation and propagation under this condition. When the direction of maximum permeability is inconsistent with the direction of maximum principal stress ($\xi = 0.5$, $\lambda > 0$), the initiation and fracture pressures of the fracturing hole slightly increase with the increasing $\lambda$, but the increasing rate obviously decreases with the increase in $\lambda$. For instance, when the $\lambda$ changes from 2 to 10, the corresponding crack initiation and fracture pressures are 5.20 and 8.5 MPa, and 5.75 and 9.25 MPa, respectively. The crack initiation and fracture pressures increase by 16.85% and 3.03% for $\lambda = 2$ compared with those of $\lambda = 1$. However, the crack initiation and fracture pressures only increase by 10.57% and 8.82% for $\lambda = 10$ compared with those of $\lambda = 2$, respectively.

③ When $\xi = 0.5$, regardless of $\lambda$, the ratio of the crack initiation pressure to the fracture pressure of fracturing holes is within the range of 53–61.18%, indicating that the ratio is less affected by the parameter $\lambda$, and that it can be used as an important indicator to predict the crack fracture pressure under the specific initiation pressure state. In addition, the AE behavior under different $\lambda$ when $\xi = 0.5$ can also be divided into three stages: the initiation stage, the fracture smooth expansion stage and the breakdown stage. The characteristic of crack length evolution and AE behavior with the calculation step is that same as that in Sections 2.2 and 2.3.

*3.5. Influence of Crack Forms on the Gas Drainage Effect*

The crack distribution form around the fracturing hole is one of the crucial factors affecting the gas drainage effect. Figure 15 shows the cloud diagram of the gas pressure distribution in the process of gas drainage under different crack forms (no crack, simple crack and complex crack). Furthermore, the change curve of cumulative gas extraction volume (CGEV) with time is also presented in Figure 16. It can be seen from Figures 15 and 16:

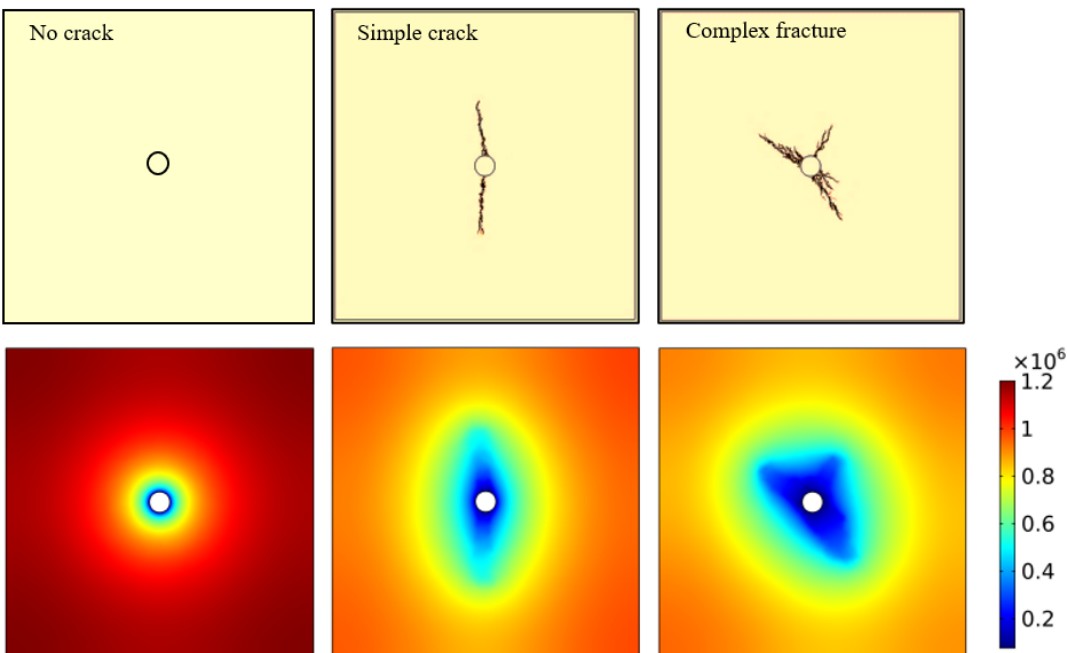

**Figure 15.** The cloud diagram of the gas pressure distribution considering different crack forms.

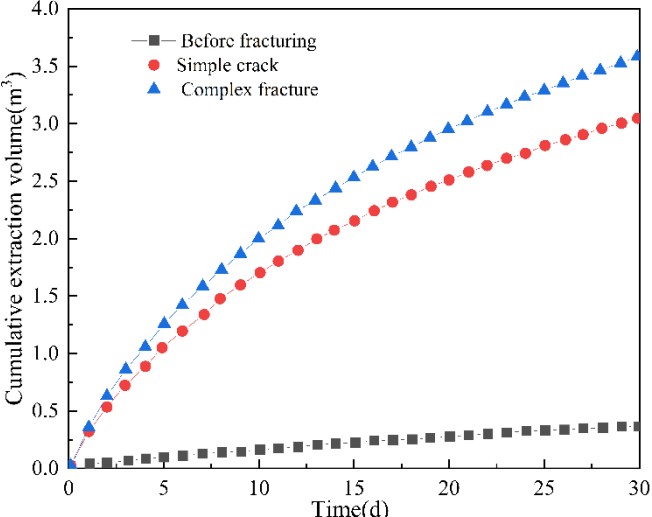

**Figure 16.** The change curve of cumulative gas extraction volume with time.

① The distribution characteristic of the gas pressure release zone (GPRZ) is consistent with the crack propagation pattern. Compared with the area distribution of GPRZ without primary cracks, the area of GPRZ after hydraulic fracturing is significantly increased. The more complex the crack distribution, the more the area of GPRZ increases.

② For any kind of crack, the curve of CGEV shows a trend of rapid increase first and then slowly increases with the gradually decreasing growth rate. Both the CGEV and growth rate of CGEV curve simultaneously show the characteristic of complex crack > simple crack > no crack. Significantly, the CGEV under complex and simple crack conditions is significantly larger than that under the condition without cracks. The above analysis indicates that the equivalent radius of gas drainage holes increases significantly after hydraulic fracturing, especially for complex cracks; the increase amplitude is more significant than that for simple cracks. The more complex the crack, the more favorable it is to gas drainage.

## 4. Discussion

### 4.1. Influence of Initial Pore Pressure on Initiation Pressure

Figure 17 shows the crack initiation pressure of fracturing holes under different initial pore pressures and calculation methods. When $\xi = 0.5$, by comparing the numerical solutions of crack initiation pressure under different initial pore pressures and the analytical solutions of crack initiation pressure based on H-W, H-F and H-BX models in this paper [36–38], it can be seen that the numerical calculation results are contrary to the results obtained by H-W and H-F models, but are highly consistent with the results predicted by the H-BX model. With the increase in initial pore pressure, the initiation pressure increases linearly. This is mainly because the H-W and H-F models ignore the influence of pore pressure gradient, and the prediction results of initiation pressure under different initial pore pressure conditions will be distorted, and perhaps even negative. The numerical calculation results and the variation trend of the initiation pressure with the initial pore pressure are close to the results obtained by the H-BX model, but slightly smaller than the results obtained by the latter, which is mainly related to the heterogeneity of coal masses. Therefore, the correctness of the model in this paper is proved again.

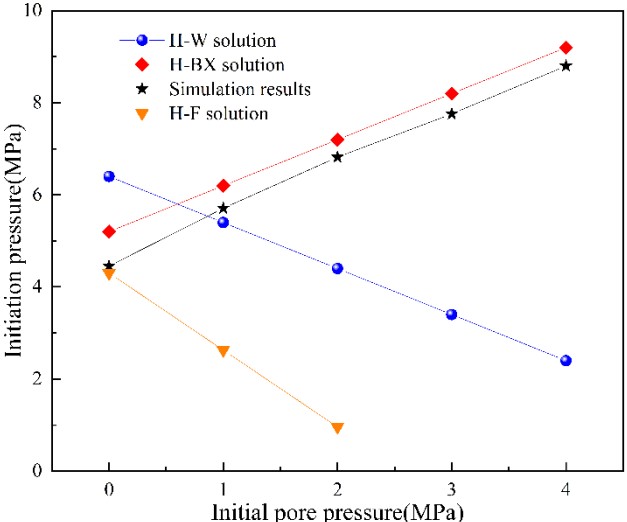

**Figure 17.** Change in crack initiation stress of fracturing holes with different initial pore pressures under various calculation methods.

In this numerical calculation, the influence of water or initial pore pressure on the surrounding rock strength attenuation effect of fracturing holes is ignored. Therefore, it is concluded that the initiation pressure increases as the initial pore pressure increases. However, if the initial pore pressure has an obvious influence on the surrounding rock strength of fracturing holes, that is, the Mohr envelope line moves downward and the Mohr–Coulomb circle moves left during the fracturing process, the peak strength of coal can be reached under the low pore pressure (see Figure 18). This means that different conclusions may be obtained.

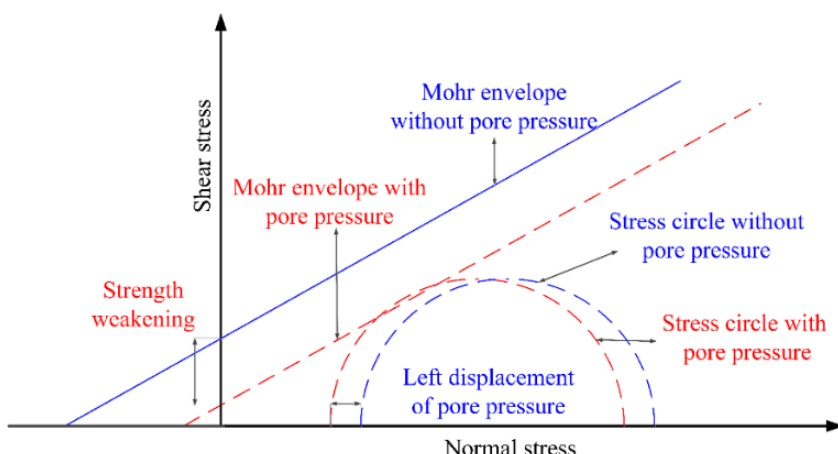

**Figure 18.** Relationship curve between shear stress and normal stress with and without pore pressure.

*4.2. Crack Propagation Complexity*

The crack propagation morphology includes length, width, angle, direction and other indexes, and its essence is the selection of dominant paths in the evolution process of crack development. If some measures are taken to narrow the dominant path of crack seepage, can the crack complexity be increased to achieve the optimal gas drainage effect? This section verifies the method through the numerical simulation. In the numerical calculation process, the crack is first prefabricated, and then the crack is blocked by graded particles to form a closed film in the seepage channel. Finally, the hydraulic fracturing process is carried out. The homogeneous ground stress condition is applied in this model. The specific numerical calculation model is shown in Figure 19.

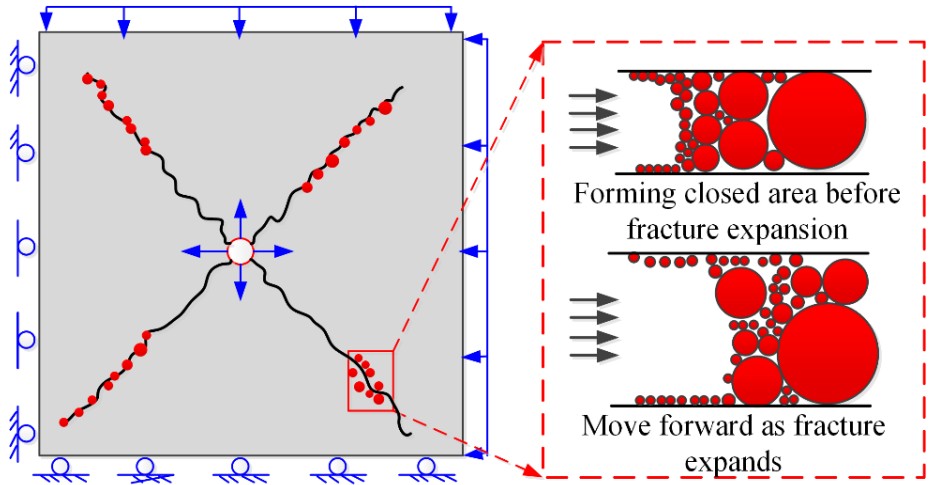

**Figure 19.** Numerical calculation model of hydraulic fracturing considering the crack sealing with special graded particles.

Figure 20 shows the crack evolution process of fracturing holes considering the particle plugging measures before and after hydraulic fracturing. It can be seen from Figure 20a that the number of dominant seepage paths increases significantly, and the path distribution becomes more complex than that without particle plugging. This is mainly because the permeability of the dominant seepage path is greatly reduced as its seepage path is blocked by grading particles. Then, the fluid leakage in the fracturing process decreases, and the pressure applied to the secondary seepage path significantly increases, which results in random crack propagation and improves the complexity of crack distribution. It is very beneficial to enhance the effect of gas extraction.

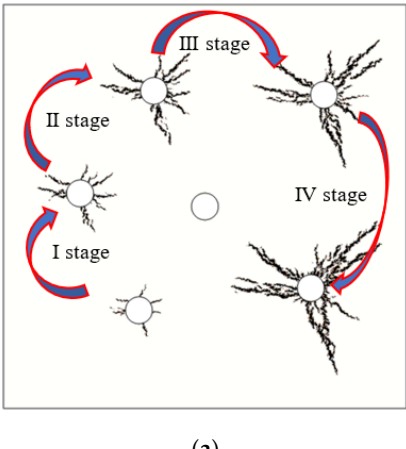
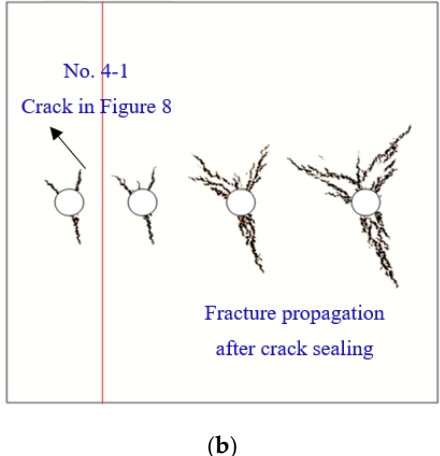

(**a**)　　　　　　　　　　　　　　　　　　　　　　　(**b**)

**Figure 20.** Crack propagation law of fracturing hole considering particle plugging measures before and after hydraulic fracturing: (**a**) particle plugging before hydraulic fracturing; (**b**) particle plugging after hydraulic fracturing.

In order to further verify the effectiveness and feasibility of this method, the number of fracturing steps was extracted as the No. 4-1 crack in Figure 8, and then the particles were used to block the crack. The simulation results indicate that the crack propagation path becomes more complex than that without particle plugging by comparison of Figures 8b and 20b. The result is basically consistent with that in Figure 20a. Hence, the feasibility of using special graded particles to seal cracks to improve the complexity of fracturing cracks is confirmed.

## 5. Conclusions

(1)　Regardless of whether the permeability is isotropic or anisotropic, the in situ stress still plays a leading role in the direction of crack propagation. Under the permeability isotropic condition, the crack initiation and fracture pressures corresponding to the non-uniform pressure are significantly lower than the above two thresholds corresponding to the uniform pressure. When the direction of maximum permeability is consistent with the direction of maximum principal stress ($\xi = 0.5$, $\lambda < 0$), the initiation and fracture pressures of fracturing holes obviously decrease with the decreasing $\lambda$, and the decreasing rate increases significantly with the decrease in $\lambda$. When the direction of maximum permeability is inconsistent with the direction of maximum principal stress ($\xi = 0.5$, $\lambda > 0$), the initiation and fracture pressures of fracturing holes slightly increase with the increasing $\lambda$, but the increasing rate obviously decreases with the increase in $\lambda$.

(2)　For any $\lambda$ or $\xi$, the crack propagation process and AE behavior of fracturing holes can be divided into three stages: the initiation stage, the fracture smooth expansion stage and the breakdown stage. In the initiation stage, the cumulative crack length and AE counts are both at a low level. However, when the fracturing process enters the fracture smooth expansion stage, the above two parameters begin to increase at a low rate. Once the fracturing process starts to enter the breakdown stage, the above two parameters increase significantly with a high growth rate, indicating that the parameters ($\lambda$ or $\xi$) have little influence on the changing characteristics of crack length and AE count with the calculation step.

(3)　The distribution characteristic of GPRZ is consistent with the crack propagation pattern. The more complex the crack distribution, the more the area of GPRZ increases. Both the CGEV and CGEV growth rate curves simultaneously show the characteristic of complex crack > simple crack > no crack. This means that the more complex the crack, the more favorable it is to gas drainage.

(4) Compared with no particle plugging measures, the number of dominant seepage paths significantly increases, and the path distribution becomes more complex, with the implementation of particle sealing measures, which is beneficial to improve the effect of gas extraction.

**Author Contributions:** Conceptualization, D.Z.; methodology, L.C.; software, Z.F.; validation, L.C., Z.F.; formal analysis, G.F.; investigation, X.W.; resources, W.Z.; data curation, N.Y.; writing—original draft preparation, L.C.; writing—review and editing, Z.F.; visualization, G.F.; supervision, X.W.; project administration, D.Z.; funding acquisition, L.C. All authors have read and agreed to the published version of the manuscript.

**Funding:** This research was funded by National Natural Science Foundation of China (52104100), China Postdoctoral Science Foundation (2021M703503) and open-ended fund of Hubei Key Laboratory for Efficient Utilization and Agglomeration of Metallurgic Mineral Resources (2020zy002).

**Institutional Review Board Statement:** Not applicable.

**Informed Consent Statement:** Not applicable.

**Data Availability Statement:** All data, models, or codes that support the findings of this study are available from the corresponding author upon reasonable request.

**Acknowledgments:** Reviewers are thanked for their insightful suggestions and comments, which improved the quality of this manuscript.

**Conflicts of Interest:** The authors declare no conflict of interest.

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
