# Peer review of "Numerical Simulation of Crack Initiation and Propagation Evolution Law of Hydraulic Fracturing Holes in Coal Seams Considering Permeability Anisotropy and Damage"

_minerals, doi:10.3390/min12040494_

Round 1
Reviewer 1 Report
This paper presents a numerical study on the crack initiation and propagation in hydraulic fracturing considering permeability anisotropy and damage. Overall, the investigation is systematic and scientifically sound. However, it has the following main problems to address:
(1) English writing. There are so many non-professional expressions throughout the paper, such as "fracture pressure", "rock initiation pressure", "fractured water", and "solve the model", which lack pre-definitions when firstly appear. Some plurals are used in the singular form, such as the subtitles in Section 2. In addition, there are some other typos and grammar errors. Therefore, the manuscript should be significantly improved.
(2) Figures and tables: Font size and type are different between figures. In some figures, such as Figure 7 and Figure 8, there is no definition for the colorbar. The unit for elastic modulus should be "GPa" rather than "Gpa". Figure 1 and Figure 13 are distorted. In Table 1, parameters are presented without definitions.
(3) Abstract: I suggest to reduce the Abstract rather than simply present all results. The parameters and abbreviations should be explained when first appear.
(4) If possible, please move some theoretical derivations to appendix so that the paper is more concise.
Reviewer 2 Report
Manuscript ID: minerals-1666138
Title: Numerical simulation on crack initiation and propagation evolution law of hydraulic fracturing holes in coal seams considering permeability anisotropy and damage
Authors: Liang Chen , Gangwei Fan , Dongsheng Zhang , Zhanglei Fan * , Xufeng Wang , Wei Zhang , Nan Yao
The authors presented the seepage stress damage coupling model, validation, and incorporation in a coupled finite element code, COMSOL.
The authors addressed the knowledge gap and novelty in this manuscript. The presentation of governing equations and constitutive laws, numerical model formulation, and validation are well documented in this manuscript.
Round 2
Reviewer 1 Report
Accept as is. The manuscript has been significantly improved.